# Random Selection Reveals Implicit Knowledge Consensus in Code Generation

**Ren-Biao Liu** [1 2]  **Xin-Ye Li** [1 2]  **Hui Sun** [1 2]  **Yali Du** [1 2]  **Jiang-Tian Xue** [1 2]  **Ming Li** [1 2]

## Abstract

Training large language models for code generation often involves selecting data from verifiable multi-solution pools, where each problem admits multiple correct implementations. Conventional studies on data selection suggest that complex selection strategies, such as diversity maximization or difficulty ranking, should outperform naive random sampling. In this work, we systematically evaluate within-problem solution selection strategies across different representation spaces, including continuous embeddings, discrete tokens, and syntactic structures, using various base language models. Instead, simple random sampling achieves consistently competitive performance across all models, exhibiting greater cross-model stability than complex methods. We interpret these results through the lens of *implicit knowledge consensus*: verified solution pools may contain representative algorithmic patterns that random sampling can preserve. Our findings suggest that practitioners should treat random sampling as a low-cost default for verifiable code-generation fine-tuning and move to complex selectors when hard-tail or constraint-focused coverage is the target.

## 1. Introduction

Code generation (Chen et al., 2021; Li et al., 2022; Roziere et al., 2023; Jiang et al., 2026) aims to automatically synthesize executable programs that satisfy given specifications. Recently, Large Language Models (LLMs), trained on vast corpora of natural language and code, have demonstrated strong capabilities for understanding and generating complex programs (Brown et al., 2020; Nijkamp et al., 2023; Guo et al., 2024; Yang et al., 2024), attracting considerable

[1]State Key Laboratory of Novel Software Technology, Nanjing University, Nanjing 210023, China [2]School of Artificial Intelligence, Nanjing University, Nanjing, China. Correspondence to: Ming Li <lim@lamda.nju.edu.cn>.

*Proceedings of the 43rd International Conference on Machine Learning*, Seoul, South Korea. PMLR 306, 2026. Copyright 2026 by the author(s).

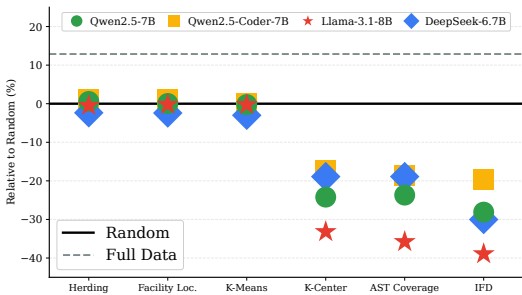

*Figure 1.* Performance comparison of seven data selection strategies averaged (arithmetic mean of absolute Pass@1 scores) across four language models. Random sampling achieves competitive average performance with no computational overhead, making it a strong baseline.

attention from both academia and industry. However, training large-scale LLMs with many learnable parameters remains highly resource-intensive (Brown et al., 2020; Kaplan et al., 2020). As a result, LLMs need to improve computational efficiency and task performance when adapting to specific downstream domains (Hoffmann et al., 2022).

Supervised Fine-Tuning (SFT) (Ouyang et al., 2022; Bai et al., 2022) offers a promising solution to enhance the specific capabilities of pre-trained LLMs for downstream tasks. Prior studies (Wei et al., 2024b; Luo et al., 2024; Gunasekar et al., 2023) suggest that high-quality training datasets for SFT can lead to significant performance gains with fewer time and resource requirements. Specifically, Zhou et al. (2023) emphasizes that carefully curated datasets can achieve competitive performance with only a fraction of the training data. This observation raises a fundamental question: *how should one curate training data to maximize the effectiveness of SFT for code generation?*

A distinctive characteristic of code datasets, particularly those derived from competitive programming platforms or verifier-backed generation pipelines, is the abundance of multiple correct solutions for each problem. Such pools also arise in RLVR-style or verifier-backed pipelines, where test-based filtering or best-of-$N$ rejection sampling can accumulate many accepted traces for the same problem before later SFT reuse. A single algorithmic challenge may admit dozens or even hundreds of valid implementations, varying in algorithmic approach, data structure choice, coding style,

and syntactic conventions. This abundance presents both an opportunity and a challenge. While diverse solutions provide rich training signals, training on the complete solution space incurs substantial computational costs and may yield diminishing returns due to redundancy.

Conventional wisdom from coreset selection and active learning suggests that diversity-maximizing strategies should yield superior generalization (Sener & Savarese, 2018; Coleman et al., 2020; Feldman, 2020). However, existing work primarily focuses on *which* question to select, rather than *how* to choose among multiple equivalent correct solutions. When faced with numerous verified solutions per problem, which accepted solutions should be retained for later fine-tuning? This question remains unexplored, yet it has significant practical implications for efficient training.

In this paper, we rigorously test the data selection in the context of fine-tuning for code generation. We use a comprehensive dataset using LeetCode problems (Xia et al., 2025). This dataset provides an opportunity to study since each problem has valid implementations with varying algorithmic approaches and coding styles. We systematically compare selection strategies ranging from naive random sampling to complex diversity-maximizing algorithms grounded in coreset selection and active learning. Our experiments across four base models yield the results shown in Figure 1: simple random sampling achieves competitive performance across all models with no computational overhead. While complex methods may occasionally outperform random sampling on specific models, they exhibit inconsistent behavior and incur significant selection costs. This finding suggests that, despite its simplicity, random selection can preserve shared algorithmic patterns across multiple correct solutions. Additional difficulty- and constraint-focused analyses further show that complex selectors become more useful when evaluation emphasizes hard-tail or constrained coverage. Our work makes the following contributions:

- **Systematic Evaluation**. We establish a rigorous experimental framework for evaluating six data selection strategies across four language models spanning multiple representation spaces.

- **Surprising Finding**. We demonstrate that random sampling achieves competitive performance across all models and exhibits greater cross-model stability than other complex methods.

- **Boundary Analysis**. We provide ablation studies and boundary analyses over distance metrics, selection budgets, difficulty, and constraints, clarifying when moving beyond random selection is useful.

The remainder of this paper is organized as follows: Section 2 formalizes the problem setting and evaluation metrics.

Section 3 describes the selection strategies in detail. Section 4 presents our main experimental results. Section 5 provides boundary analyses clarifying the scope of our findings. Section 6 reviews related work on code generation and data selection. Section 7 concludes findings.

## 2. Preliminaries

This section formalizes the data selection problem for code generation, where each programming problem admits multiple correct solutions. We introduce the evaluation metrics and computational considerations that motivate our study.

### 2.1. Problem Formulation

Let $\mathcal{D} = \{(p_i, \mathcal{S}_i)\}_{i=1}^N$ denote a SFT dataset, where each instance comprises a problem description $p_i$ and an associated set of verified correct solutions $\mathcal{S}_i = \{s_i^1, s_i^2, \ldots, s_i^{m_i}\}$. Here, $N$ denotes the total number of problems and $m_i = |\mathcal{S}_i|$ represents the cardinality of the solution set for problem $p_i$. In typical SFT datasets derived from competitive programming platforms, each problem admits multiple valid implementations, each employing a different algorithmic approach.

**Data Selection.** Given a per-problem selection budget $k \in \mathbb{Z}^+$, the data selection task seeks a mapping $\phi : \mathcal{S}_i \to \mathcal{S}_i'$ that produces a subset $\mathcal{S}_i' \subseteq \mathcal{S}_i$ satisfying $|\mathcal{S}_i'| \leq \min(k, m_i)$. The resulting selected dataset is denoted $\mathcal{D}' = \{(p_i, \mathcal{S}_i')\}_{i=1}^N$. The optimization objective can be formally stated as:

$$\phi^* = \arg\max_\phi \ \mathcal{M}\big(f_\theta(\mathcal{D}')\big), \tag{1}$$

where $f_\theta(\mathcal{D}')$ denotes a language model fine-tuned on the selected dataset $\mathcal{D}'$, and $\mathcal{M}(\cdot)$ represents a downstream evaluation metric (e.g., Pass@$k$). In practice, this objective is intractable to optimize directly because it requires training a model for each candidate selection (Coleman et al., 2020); therefore, selection strategies employ proxy objectives instead.

### 2.2. Evaluation Metrics

We adopt the Pass@$k$ metric (Chen et al., 2021; Kulal et al., 2019), which has become the standard evaluation measure for code generation systems. Given a problem $p$, we generate $n$ candidate solutions from the model and define:

$$\text{Pass@}k = \mathbb{E}_{p \sim \mathcal{P}_{\text{test}}} \left[ 1 - \frac{\binom{n-c}{k}}{\binom{n}{k}} \right], \tag{2}$$

where $c$ denotes the number of correct solutions among the $n$ samples that pass all test cases, and $\binom{n}{k}$ denotes the binomial coefficient. It is an unbiased estimator of the probability that at least one of $k$ solutions is correct (Chen et al., 2021).

**Evaluation Protocol.** For the training set, we use greedy decoding (temperature $\tau = 0$) with a single sample per problem to compute Pass@1. For the test set, we generate $n = 10$ samples per problem with temperature $\tau = 1.0$ and report Pass@$k$ for $k \in \{1, 2, 3, 5, 10\}$. Pass@1 reflects single-attempt accuracy, while higher $k$ values measure the model's ability in test time scaling.

## 2.3. Supervised Fine-Tuning

SFT adapts a base model to downstream tasks (Ouyang et al., 2022; Bai et al., 2022). Given a dataset $\mathcal{D}' = \{(p_i, s_i)\}_{i=1}^{|\mathcal{D}'|}$ where $p_i$ is the problem description and $s_i$ is a solution, the SFT objective minimizes:

$$\mathcal{L}(\theta) = -\sum_{i=1}^{|\mathcal{D}'|} \sum_{t=1}^{|s_i|} \log P_\theta(s_i^{(t)} \mid p_i, s_i^{(<t)}), \qquad (3)$$

where $s_i^{(t)}$ denotes the $t$-th token of solution $s_i$, and $s_i^{(<t)}$ represents all preceding tokens. This autoregressive formulation trains the model to predict each token given the problem and the previously generated tokens.

**Computational Considerations.** The computational cost of SFT scales linearly with the dataset size $|\mathcal{D}'|$, making data selection critical for large-scale training. By selecting a subset of solutions per problem (budget $k$), we reduce training cost by a factor of approximately $\bar{m}/k$, where $\bar{m}$ is the average number of solutions per problem. This substantial reduction in training compute motivates a systematic study of effective data-selection strategies for code generation.

## 3. Data Selection Methods

### 3.1. Dataset Statistics and Selection Objective

**Data Source.** We study data selection on a curated Leet-Code dataset (Xia et al., 2025; Morales, 2024) comprising $N = 2,641$ programming problems. Each problem $p_i$ is associated with a solution set $\mathcal{S}_i = \{s_i^1, s_i^2, \ldots, s_i^{m_i}\}$ of verified correct solutions. The complete raw dataset contains over $500K$ Python solutions, with the number of valid solutions per problem varying widely, from 1 to over $5K$.

**Data Preprocessing.** To ensure computational feasibility and reduce redundancy, we apply Locality Sensitive Hashing (LSH) (Broder, 1997; Indyk & Motwani, 1998) to remove near-duplicates. Each solution is tokenized into Python tokens and converted to 3-gram shingles, from which we compute 256-permutation MinHash signatures. Solutions with estimated Jaccard similarity exceeding 0.85 are grouped via indexing and union-find clustering. From each cluster, we retain only the most representative solution with the highest average similarity to other members. After this

preprocessing, the cardinality $m_i = |\mathcal{S}_i|$ for each problem is capped at 100, ensuring that selection strategies operate on a pool of meaningfully distinct solutions while maintaining computational tractability. This shared preprocessing is treated as corpus cleaning rather than as a competing selection strategy.

**Sources of Solution Multiplicity.** The abundance of solutions per problem arises from three factors. *Algorithmic diversity*: problems admit solutions with fundamentally different algorithmic approaches, each with distinct time-space complexity tradeoffs. *Implementation variance*: even for identical algorithms, developers employ different coding styles, variable naming conventions, and control flow structures. *Community heterogeneity*: competitive programming platforms accumulate solutions from thousands of users with diverse backgrounds and programming preferences.

**Proxy Objective.** For each problem $p_i$ with $m_i$ solutions, we seek a subset $\mathcal{S}_i' \subseteq \mathcal{S}_i$ of size $|\mathcal{S}_i'| \leq \min(k, m_i)$ that maximizes downstream model performance (Equation 1). As discussed in Section 2, direct optimization of this objective is intractable. Practical selection strategies instead optimize *proxy objectives* $\mathcal{J}(\mathcal{S}_i')$ that encode prior beliefs about what constitutes an effective training subset. The central question is whether these proxy objectives, typically designed to promote diversity or coverage, correlate with the true downstream metric.

### 3.2. Baseline and Upper Bound

**Uniform Random Sampling.** The simplest approach is to sample solutions uniformly at random without replacement. For each problem $p_i$ with solution set $\mathcal{S}_i$, we draw $k_i = \min(k, m_i)$ solutions uniformly at random. Random sampling requires no proxy objective optimization and serves as a natural baseline for comparison.

**Full Dataset Training.** As an upper bound for selection, we evaluate training on the complete dataset without any selection ($|\mathcal{S}_i'| = m_i$ for all problems). This approach uses all deduplicated Python solution instances for SFT.

### 3.3. Core Selection Strategies

**K-Center Greedy.** K-Center (Gonzalez, 1985; Sener & Savarese, 2018) minimizes the maximum distance from any point to its nearest selected representative, ensuring uniform coverage of the solution space:

$$\mathcal{S}_i'^* = \arg \min_{\substack{\mathcal{S}_i' \subseteq \mathcal{S}_i \\ |\mathcal{S}_i'| \leq k}} \max_{s \in \mathcal{S}_i} \min_{s' \in \mathcal{S}_i'} \delta(s, s'). \qquad (4)$$

The greedy algorithm iteratively selects the point farthest from the current selection, yielding a 2-approximation to the

*Table 1.* Formal comparison of six core data selection strategies. Each method is characterized by its representation space $\mathcal{X}$, optimization objective $\mathcal{J}(\mathcal{S}'_i)$, theoretical approximation guarantee, and computational complexity.

| Method | Space $\mathcal{X}$ | Objective $\mathcal{J}(\mathcal{S}'_i)$ | Guarantee | Time | Space |
|---|---|---|---|---|---|
| K-Center | $\mathcal{X}$ | $\min\limits_{\mathcal{S}'_i} \max\limits_{s \in \mathcal{S}_i} \min\limits_{s' \in \mathcal{S}'_i} \delta(s, s')$ | 2-approximation | $O(km_i C_\delta)$ | $O(m_i)$ |
| Facility Location | $\mathcal{X}$ | $\max\limits_{\mathcal{S}'_i} \sum\limits_{s \in \mathcal{S}_i} \max\limits_{s' \in \mathcal{S}'_i} \mathrm{sim}(s, s')$ | $(1-1/e)$-approx | $O(km_i C_{\mathrm{sim}})$ | $O(m_i)$ |
| K-Means | $\mathbb{R}^d$ | $\min\limits_{\{C_j\}} \sum\limits_{j=1}^{k} \sum\limits_{s \in C_j} \|\mathbf{e}_s - \boldsymbol{\mu}_j\|_2^2$ | Local optimum | $O(m_i k d T)$ | $O(m_i d)$ |
| AST Coverage | $2^{\mathcal{U}}$ | $\max\limits_{\mathcal{S}'_i} \left\| \bigcup\limits_{s \in \mathcal{S}'_i} \mathcal{P}_s \right\|$ | $(1-1/e)$-approx | $O(km_i |\mathcal{U}|)$ | $O(|\mathcal{U}|)$ |
| Kernel Herding | $\mathbb{R}^d$ | $\min\limits_{\mathcal{S}'_i} \mathrm{MMD}^2(\mathcal{S}'_i, \mathcal{S}_i)$ | — | $O(km_i d)$ | $O(m_i d)$ |
| IFD Ranking | $\mathbb{R}$ | $\max\limits_{\mathcal{S}'_i} \sum\limits_{s \in \mathcal{S}'_i} \mathrm{IFD}(s, p_i)$ | — | $O(m_i L)$ | $O(m_i)$ |

*Notation:* $m_i = |\mathcal{S}_i|$ (solutions per problem), $d = 768$ (embedding dimension), $L$ (avg. token length), $T$ (iterations), $\mathcal{U}$ (set of unique AST subtree patterns observed in the input solutions; $|\mathcal{U}| \approx 50,000$ in our dataset), $\mathcal{X}$ (representation space), $C_\delta$ (distance computation cost; $O(L^2)$ for Levenshtein, $O(L)$ for Jaccard), $C_{\mathrm{sim}}$ (similarity computation cost; $O(d)$ for cosine, $O(L)$ for Jaccard). For IFD, $L$ denotes the forward pass cost proportional to sequence length.

optimal solution. We use token-level Levenshtein distance as the default distance metric.

**Facility Location.** Facility Location (Cornuéjols et al., 1990) maximizes total similarity between all points and their nearest selected representatives, favoring solutions that collectively represent the entire pool:

$$\mathcal{S}'^*_i = \arg \max_{\substack{\mathcal{S}'_i \subseteq \mathcal{S}_i \\ |\mathcal{S}'_i| \le k}} \sum_{s \in \mathcal{S}_i} \max_{s' \in \mathcal{S}'_i} \mathrm{sim}(s, s'). \quad (5)$$

Unlike K-Center's minimax objective, this sum-based formulation achieves $(1-1/e)$-approximation via greedy selection due to submodularity (Nemhauser et al., 1978). We use cosine similarity on embeddings output by Code-BERT (Feng et al., 2020) as the default similarity metric.

**K-Means Clustering.** K-Means (Lloyd, 1982) partitions solutions into $k$ clusters by minimizing within-cluster variance, then selects representative solutions from each cluster:

$$\{C_j^*\}_{j=1}^k = \arg \min_{\{C_j\}_{j=1}^k} \sum_{j=1}^{k} \sum_{s \in C_j} \|\mathbf{e}_s - \boldsymbol{\mu}_j\|_2^2, \quad (6)$$

where $\boldsymbol{\mu}_j = \frac{1}{|C_j|} \sum_{s \in C_j} \mathbf{e}_s$ is the centroid of cluster $C_j$, subject to $\bigcup_{j=1}^{k} C_j = \mathcal{S}_i$ and $C_j \cap C_{j'} = \emptyset$ for all $j \ne j'$. We select the nearest solution to each centroid with k-means++ initialization (Arthur & Vassilvitskii, 2007).

**AST Coverage.** AST Coverage (Du et al., 2025) maximizes coverage of distinct abstract syntax tree patterns, capturing structural diversity in code implementations rather

than surface-level textual similarity:

$$\mathcal{S}'^*_i = \arg \max_{\substack{\mathcal{S}'_i \subseteq \mathcal{S}_i \\ |\mathcal{S}'_i| \le k}} \left| \bigcup_{s \in \mathcal{S}'} \mathcal{P}_s \right|, \quad (7)$$

where $\mathcal{P}_s$ contains AST subtree fingerprints extracted via tree-sitter parsing; each subtree is hashed after abstracting variable names to capture structural patterns.

**Kernel Herding.** Kernel Herding (Chen et al., 2010) minimizes the maximum mean (MMD) discrepancy between the selected subset and the full solution distribution in embedding space. With a linear kernel, MMD reduces to the squared distance between mean embeddings:

$$\mathcal{S}'^*_i = \arg \min_{\substack{\mathcal{S}'_i \subseteq \mathcal{S}_i \\ |\mathcal{S}'_i| = k}} \left\| \frac{1}{k} \sum_{s \in \mathcal{S}'_i} \mathbf{e}_s - \frac{1}{m_i} \sum_{s \in \mathcal{S}_i} \mathbf{e}_s \right\|_2^2. \quad (8)$$

The greedy algorithm iteratively selects solutions that minimize the distance between the subset and the population means, prioritizing distributional matching over diversity.

**Instruction Following Difficulty Ranking.** IFD Ranking (Li et al., 2024) selects solutions with the greatest Instruction Following Difficulty (IFD), prioritizing samples where the context significantly influences the solution:

$$\mathrm{IFD}(s, p_i) = \frac{\mathcal{L}(s \mid p_i)}{\mathcal{L}(s)}, \quad (9)$$

where $\mathcal{L}(s \mid p_i)$ and $\mathcal{L}(s)$ are conditional and unconditional losses computed using DeepSeek-Coder-6.7B-Base. We use a single proxy model to compute IFD scores for all target models, as the ranking reflects intrinsic sample difficulty rather than model-specific properties.

*Table 2.* Comprehensive results (%) after training completion across all models and methods. We report Pass@1 on the training set (Train) and Pass@k for $k \in \{1, 2, 3, 5, 10\}$ on the test set (Test). For each model and metric, **best** results are in bold, and second-best results are underlined (comparing across Random and six core selection methods; Full Dataset excluded from ranking). Full Dataset trained for $6\times$ longer (epoch-matched, not compute-matched).

| | Qwen2.5-7B | | | | | | Qwen2.5-Coder-7B | | | | | |
|---|---|---|---|---|---|---|---|---|---|---|---|---|
| **Strategy** | Train | Test | | | | | Train | Test | | | | |
| | Pass@1 | Pass@1 | Pass@2 | Pass@3 | Pass@5 | Pass@10 | Pass@1 | Pass@1 | Pass@2 | Pass@3 | Pass@5 | Pass@10 |
| Random | 65.20 | **6.32** | **9.83** | **12.17** | **15.26** | **19.30** | 69.63 | 5.57 | 8.88 | 11.24 | 14.47 | 19.30 |
| K-Center | 49.41 | 3.90 | 6.36 | 8.25 | 11.21 | 16.23 | 57.63 | 5.26 | 8.60 | 11.07 | 14.55 | 19.30 |
| Facility Location | 65.24 | 4.61 | 7.51 | 9.69 | 12.98 | 17.98 | **70.43** | 6.05 | 9.51 | 11.90 | 15.29 | **20.61** |
| AST Coverage | 49.75 | 3.73 | 6.03 | 7.73 | 10.26 | 14.47 | 56.68 | 4.96 | 7.78 | 9.69 | 12.40 | 17.11 |
| K-Means | 65.05 | 4.87 | 7.78 | 9.85 | 12.72 | 16.67 | 69.63 | **6.14** | **9.68** | **12.23** | **15.69** | 20.18 |
| Kernel Herding | **65.58** | 6.23 | 9.48 | 11.69 | 14.63 | 18.42 | 70.39 | 5.88 | 9.07 | 11.25 | 14.30 | 19.30 |
| IFD Ranking | 46.88 | 3.42 | 5.48 | 6.94 | 8.94 | 11.40 | 56.00 | 4.78 | 7.48 | 9.42 | 12.30 | 16.67 |
| Full Dataset | 72.02 | 6.14 | 9.88 | 12.46 | 15.86 | 20.18 | 74.14 | 6.54 | 10.14 | 12.65 | 16.03 | 20.61 |
| | Llama-3.1-8B | | | | | | DeepSeek-Coder-6.7B | | | | | |
| **Strategy** | Train | Test | | | | | Train | Test | | | | |
| | Pass@1 | Pass@1 | Pass@2 | Pass@3 | Pass@5 | Pass@10 | Pass@1 | Pass@1 | Pass@2 | Pass@3 | Pass@5 | Pass@10 |
| Random | **54.18** | **4.04** | **6.64** | **8.46** | 10.85 | 14.04 | 64.33 | 3.68 | 6.02 | 7.62 | 9.74 | 12.72 |
| K-Center | 36.20 | 2.63 | 4.54 | 6.01 | 8.20 | 11.84 | 52.18 | 2.50 | 4.43 | 5.95 | 8.15 | 11.40 |
| Facility Location | 54.07 | 3.90 | 6.35 | 8.15 | 10.76 | 14.91 | 62.78 | 4.08 | 6.74 | 8.67 | 11.43 | 15.79 |
| AST Coverage | 34.80 | 3.03 | 5.01 | 6.45 | 8.48 | 11.40 | 52.18 | 2.59 | 4.45 | 5.86 | 7.87 | 10.96 |
| K-Means | 54.03 | 3.60 | 6.27 | 8.32 | **11.31** | **16.23** | 62.40 | 3.95 | 6.31 | 8.04 | 10.72 | 15.35 |
| Kernel Herding | 53.92 | 3.55 | 5.95 | 7.66 | 10.07 | 14.04 | 62.82 | **4.56** | **7.21** | **9.27** | **12.45** | **17.54** |
| IFD Ranking | 33.09 | 2.98 | 4.88 | 6.22 | 8.06 | 10.96 | 45.02 | 2.50 | 4.30 | 5.66 | 7.60 | 10.09 |
| Full Dataset | 67.21 | 3.68 | 6.09 | 7.86 | 10.46 | 14.47 | 71.11 | 3.95 | 6.18 | 7.72 | 9.96 | 13.60 |

### 3.4. Summary and Comparison

We have introduced six core data selection strategies and assumption-free random sampling to select an optimal subset from the entire dataset. Each strategy optimizes a distinct proxy objective, from geometric diversity to distributional matching. Table 1 provides a comparison along theoretical and computational dimensions.

## 4. Experiments

### 4.1. Experiment Setup

**Dataset and Budget.** We use the LeetCode dataset described in Section 3.1, comprising $N = 2,869$ programming problems with over $150K$ verified solution instances collected from competitive programming submissions. We partition the dataset into 2,641 training problems and 228 held-out test problems by contest date, ensuring no overlap in problem content. With a per-problem selection budget of $k = 11$ solutions, each strategy yields $\sim 25,600$ training samples. All methods use a batch size of 128 and train for 3 epochs; selection methods complete in 600 steps, while full-data training extends to 3,600 steps.

**Models and Training.** We evaluate on four models spanning different architectures: Qwen2.5-7B (Yang et al.,

2024), Qwen2.5-Coder-7B (Hui et al., 2024), Llama-3.1-8B (Grattafiori et al., 2024), and DeepSeek-Coder-6.7B (Guo et al., 2024). Training uses AdamW (Loshchilov & Hutter, 2019) with learning rate $1 \times 10^{-5}$, weight decay 0.01, linear warmup over 10% of steps, and cosine decay. All experiments use bfloat16 precision on NVIDIA A100 GPUs with 80GB of memory, and each 7B model training run takes approximately 64 GPU-hours.

### 4.2. Main Results

**Performance Patterns.** We test whether complex selection methods outperform random sampling. Table 2 presents results across four models and seven selection methods. Random achieves the highest Test Pass@1 on half of the evaluated models, and no single method consistently outperforms Random across all architectures. The best-performing method varies by model, as K-Means performs best on Qwen2.5-Coder-7B while Kernel Herding performs best on DeepSeek-Coder-6.7B. This variation highlights the model dependence of complex selection strategies, which may overfit to specific architectural characteristics or training dynamics. In contrast, Random remains a strong low-cost reference rather than a uniformly optimal selector.

**Comparison with Full Data.** Remarkably, on Qwen2.5-7B and Llama-3.1-8B, Random *outperforms* Full Dataset

*Table 3.* Statistical significance analysis: Mean $\pm$ standard deviation (%) over 5 seeds. We report Pass@1 on the training set and Pass@10 on the test set.

| Method | Train Pass@1 | Test Pass@10 |
|---|---|---|
| Random | $65.45 \pm 0.43$ | $18.07 \pm 2.31$ |
| K-Center | $49.31 \pm 0.57$ | $14.82 \pm 1.68$ |
| Facility Location | $65.01 \pm 0.76$ | $17.37 \pm 1.01$ |
| K-Means | $65.26 \pm 0.50$ | $17.19 \pm 1.00$ |
| AST Coverage | $48.63 \pm 0.71$ | $14.82 \pm 1.14$ |
| Kernel Herding | $64.78 \pm 0.66$ | $17.54 \pm 0.93$ |
| IFD Ranking | $46.86 \pm 0.41$ | $14.21 \pm 1.90$ |

despite using only a small fraction of the data and compute. Similarly, on DeepSeek-Coder-6.7B, Kernel Herding, Facility Location, and K-Means all match or exceed the full-data baseline. These findings demonstrate that strategic data selection can achieve comparable or superior performance while dramatically reducing training costs, challenging the assumption that more data invariably leads to better models. The strong performance of Random in this comparison suggests that representative accepted traces can be sufficient, while more complex objectives may help when they match the evaluation target.

### 4.3. Statistical Significance

**Significance Tests.** To validate the robustness of our findings, we conduct experiments across five random seeds on Qwen2.5-7B, where Random exhibits clear performance advantages over most selection methods. Table 3 reports the mean and standard deviation for Train Pass@1 and Test Pass@10. This multi-seed evaluation enables rigorous statistical comparison, allowing us to assess not only point estimates but also the reliability of each selection strategy. Such analysis provides confidence that the observed differences reflect genuine methodological effects rather than random variation arising from initialization or data ordering.

**Variance Analysis.** Random achieves the highest mean Test Pass@10 across five seeds, exceeding each evaluated complex selection method in mean Pass@10. While Random exhibits relatively larger variance than Kernel Herding and Facility Location, its mean performance remains higher in this setting. The result indicates that Random's effectiveness is a robust characteristic of the sampling strategy rather than an artifact of favorable initialization. Random significantly outperforms weaker methods, such as K-Center, AST Coverage, and IFD Ranking, while maintaining competitive performance relative to methods with lower variance.

### 4.4. Scaling to Larger Models

**Larger-Scale Evaluation.** To test whether our findings hold at a larger scale, we evaluate all seven methods on

*Table 4.* Scaling to larger models: Performance (%) of Qwen2.5-32B (600 steps). We report Pass@1 on the training set and Pass@1, Pass@10 on the test set. **Best** results are in bold and second-best results are underlined.

| | Train | Test | |
|---|---|---|---|
| **Strategy** | Pass@1 | Pass@1 | Pass@10 |
| Random | **79.14** | **13.82** | 31.58 |
| K-Center | 66.04 | 11.49 | 32.46 |
| Facility Location | 78.46 | 12.41 | **35.53** |
| AST Coverage | 65.69 | 10.96 | 32.46 |
| K-Means | 78.91 | 13.55 | 34.65 |
| Kernel Herding | 79.10 | 13.11 | 34.21 |
| IFD Ranking | 65.20 | 11.10 | 30.26 |

Qwen2.5-32B, a model with substantially greater capacity. As shown in Table 4, the performance hierarchy observed at more minor scales largely persists for Test Pass@1. Random achieves the best Test Pass@1, closely followed by K-Means, while K-Center, AST Coverage, and IFD Ranking continue to lag. Notably, for Test Pass@10, Facility Location, K-Means, and Kernel Herding all outperform Random at this scale, suggesting that the optimal selection strategy may depend on evaluation.

**Performance Scaling.** Scaling from 7B to 32B parameters yields substantial performance improvements across all methods, and Random's Test Pass@1 more than doubles compared to its 7B counterpart. This scaling benefit does not erase the gap between Random and the weakest proxy objectives, but the Pass@10 results reveal a metric-dependent tradeoff. Together with our cross-architecture results, these findings establish Random as a robust baseline while motivating the boundary analyses below.

*Table 5.* Constraint-focused performance (%) on 59 tagged Medium/Hard problems.

| Method | Pass@1 | Pass@10 |
|---|---|---|
| Random | **1.86** | 6.78 |
| K-Means | 1.69 | **8.47** |
| Facility Location | 1.19 | **8.47** |
| Kernel Herding | 1.36 | 6.78 |

### 4.5. Beyond Random Selection

**Constraint-Focused Evaluation.** Table 5 asks when moving beyond Random helps on 59 tagged Medium/Hard problems. This narrow subset removes Easy problems and keeps explicit algorithmic constraints, so the evaluation rewards more than the most typical accepted implementation. Random gives the best Pass@1, whereas K-Means and Facility Location obtain stronger Pass@10. This split suggests that Random remains preferable for a single representative solution, while complex selectors help when constrained

evaluations reward multiple plausible routes.

**Performance by Difficulty.** We stratify Qwen2.5-7B evaluation by LeetCode difficulty to separate the representative bulk from the hardest tail. In Table 6, Random is strongest on Easy problems and best on Medium Pass@10, while Kernel Herding overtakes it on Hard problems. The pattern separates average-case representativeness from hard-tail coverage: Easy and Medium problems still favor representative accepted traces, whereas Hard problems leave more room for selectors that search less typical regions. Thus Random remains the default, with complex selectors adding coverage for hard or constrained cases rather than replacing Random uniformly.

*Table 6.* Qwen2.5-7B performance (%) by LeetCode difficulty. Easy/Medium/Hard contain 48/101/79 tasks.

| Subset | Method | Pass@1 | Pass@10 |
|---|---|---|---|
| Easy | Random | **22.71** | **60.42** |
| | K-Means | 15.21 | 52.08 |
| | Facility Location | 16.25 | 58.33 |
| | Kernel Herding | 20.62 | 56.25 |
| Medium | Random | 2.77 | **10.89** |
| | K-Means | 2.67 | 9.90 |
| | Facility Location | 2.08 | 8.91 |
| | Kernel Herding | **2.87** | 9.90 |
| Hard | Random | 0.89 | 5.06 |
| | K-Means | 1.39 | 3.80 |
| | Facility Location | 0.76 | 5.06 |
| | Kernel Herding | **1.77** | **6.33** |

**Difficulty Stratification.** The difficulty split raises a second question: are hard-tail gains connected to the structure of the solution pools? Table 7 stratifies diversity proxies by difficulty, focusing on the axis that is most relevant to the beyond-random setting. Harder problems have larger token-set, edit, and syntax diversity, matching the performance drop. This difficulty-facing view avoids repeating the broader correlation table in Section 5.4 while showing how the same signal manifests across Easy, Medium, and Hard subsets.

*Table 7.* Solution-pool diversity and Random Pass@1 by LeetCode difficulty. Easy/Medium/Hard contain 48/101/79 tasks.

| Subset | Jaccard | Lev. | Syntax | Pass@1 |
|---|---|---|---|---|
| Easy | 0.42 | 88.29 | 0.69 | 23.08 |
| Medium | 0.48 | 179.86 | 0.75 | 2.82 |
| Hard | 0.51 | 229.00 | 0.78 | 0.89 |

**Case-Level Comparison.** Table 8 compares cases solved by both selectors, missed by both, Random-only successes, and complex-only successes. The largest error group is the shared failure set, whose higher difficulty and syntax

diversity indicate that many remaining errors reflect problem difficulty. Random-only and complex-only wins are smaller and occupy nearby boundary cases, suggesting that the methods trade individual problems instead of solving distinct regimes. Overall, Random remains the representative default, whereas diversity- or matching-based methods are targeted tools for difficult tails and constrained evaluations.

*Table 8.* Case-level outcomes for Random and complex selectors. Difficulty maps Easy/Medium/Hard to 1/2/3 and reports the group average; E/M/H reports Easy, Medium, and Hard counts.

| Outcome | Count | Difficulty | Syntax | E/M/H |
|---|---|---|---|---|
| Both success | 38 | 1.39 | 0.69 | 27/7/4 |
| Both failure | 167 | 2.37 | 0.76 | 11/83/73 |
| Random only | 6 | 1.67 | 0.73 | 2/4/0 |
| Complex only | 17 | 1.65 | 0.73 | 8/7/2 |

## 5. Discussion

### 5.1. Robustness Analysis

**Learning Rate Sensitivity.** We verify that the competitive performance of random sampling is robust to hyperparameter choices by training Qwen2.5-7B with learning rates spanning two orders of magnitude. As shown in Table 9, random sampling maintains reasonable Test Pass@1 across a range of learning rates, with optimal results at moderate values. While absolute performance varies with learning rate as expected, random sampling does not require extensive hyperparameter tuning to achieve competitive results, reducing the computational cost of hyperparameter search.

*Table 9.* Learning rate sensitivity analysis for random selection. We report Pass@1 on the training set and Pass@1, Pass@10 on the test set.

| | Train | Test | |
|---|---|---|---|
| **LR** | Pass@1 | Pass@1 | Pass@10 |
| $1e-6$ | 56.57 | 3.42 | 14.04 |
| $2e-6$ | 60.17 | 4.08 | 17.54 |
| $5e-6$ | 63.57 | 4.61 | 16.67 |
| $1e-5$ | 65.20 | 6.32 | 19.30 |
| $2e-5$ | 66.72 | 5.75 | 18.86 |
| $5e-5$ | 71.49 | 5.35 | 17.54 |

**Seed Stability.** To verify that the competitiveness of random sampling is not dependent on fortuitous seed choices, we train with multiple random seeds, each selecting a distinct subset of solutions. As shown in Table 10, Train Pass@1 remains highly stable across seeds while Test Pass@1 shows somewhat larger variation, as expected given the smaller test set size, indicating that different random samples yield similar performance levels. This stability suggests that the heterogeneous solution pools contain sufficient

inherent diversity, and different random samples can capture representative training signals without requiring seed selection or multiple runs to identify a favorable subset.

*Table 10.* Stability of random selection across five random seeds. We report Pass@1 on training set and Pass@1, Pass@10 on test.

|  | Train | Test |  |
| --- | --- | --- | --- |
| **Seed** | Pass@1 | Pass@1 | Pass@10 |
| 347 | 65.20 | 6.32 | 19.30 |
| 348 | 65.20 | 5.39 | 18.86 |
| 349 | 66.34 | 5.04 | 16.67 |
| 350 | 65.77 | 4.39 | 16.67 |
| 351 | 64.94 | 5.18 | 18.42 |
| Avg. | $65.49 \pm 0.56$ | $5.26 \pm 0.70$ | $17.98 \pm 1.24$ |

## 5.2. Distance Metrics for K-Center

We evaluate K-Center across four distance functions while holding other parameters constant, including selection budget, learning rate, and training duration.

*Table 11.* K-Center ablation: Train Pass@1 (%) with different distance metrics. Cosine distance yields higher performance than surface-level metrics, yet all variants underperform Random.

|  | **Train Pass@1** | | | |
| --- | --- | --- | --- | --- |
| **Model** | **Jaccard** | **Lev.** | **Syntax** | **Cosine** |
| Qwen2.5 | 51.50 | 49.41 | 50.55 | 58.69 |
| Qwen2.5-Coder | 58.99 | 57.63 | 56.99 | 64.26 |
| Llama-3.1 | 38.51 | 36.20 | 37.30 | 46.88 |
| DeepSeek-Coder | 52.37 | 52.18 | 52.67 | 58.27 |

As shown in Table 11, Cosine distance yields higher Train Pass@1 than other metrics within the K-Center framework across all models, likely because semantic embeddings better capture functional similarity between code snippets than surface-level text features. However, this advantage does not change the overall conclusion that K-Center variants underperform Random sampling in Train Pass@1 across all models regardless of the distance metric employed. This finding suggests that the characteristic behavior of K-Center in selecting boundary points due to its minimax objective is a fundamental limitation that prioritizes covering the extremes of the solution space rather than selecting representative examples that support generalization.

## 5.3. Selection Budget

We vary the selection budget via random sampling on Qwen2.5-7B to examine the data quantity-generalization trade-off. Each budget level determines the number of solutions per problem, with the total training set size scaling.

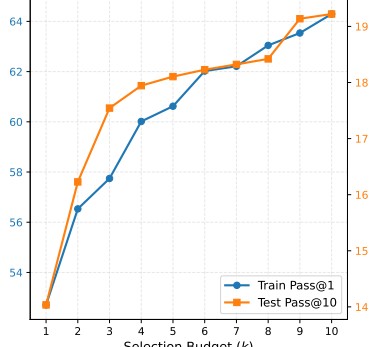

*Figure 2.* Effect of selection budget $k$ on performance. Both metrics exhibit an overall upward trend, with diminishing marginal returns at larger budgets.

As shown in Figure 2, both Train Pass@1 and Test Pass@10 exhibit a generally upward trend as budget increases. The growth rate is steepest at small budgets and gradually diminishes, suggesting that the marginal benefit of additional solutions declines as the training set size increases. This pattern indicates that a moderate selection budget suffices to capture the essential training signal, while larger budgets provide only incremental gains at higher cost.

## 5.4. Diversity and Problem Difficulty

We analyze correlations between solution diversity metrics and Pass@k performance on random sampling results. We compute pairwise distances across solutions for each problem and examine both Pearson and Spearman correlations.

*Table 12.* Correlation between diversity metrics and Pass@k performance on Random selection results. Both Pearson ($r$) and Spearman ($\rho$) show significant *negative* correlations ($p < 0.001$), indicating that higher diversity within the solution pool is associated with lower Pass@k rates. This suggests that problems with more diverse solutions are inherently more difficult.

|  | **Pass@1** | | **Pass@10** | |
| --- | --- | --- | --- | --- |
| **Metric** | $r$ | $\rho$ | $r$ | $\rho$ |
| Levenshtein | -0.447 | -0.457 | -0.430 | -0.437 |
| Jaccard | -0.423 | -0.386 | -0.394 | -0.374 |
| Syntax | -0.471 | -0.445 | -0.445 | -0.433 |

As shown in Table 12, all three diversity metrics exhibit statistically significant negative correlations with Pass@k, indicating that problems whose solution pools display greater diversity tend to have lower Pass@k rates. This inverse relationship suggests that solution diversity is associated with problem difficulty, since more challenging problems naturally admit a wider variety of solution approaches and are inherently more complex for models to solve. This finding is consistent with the view that diversity-based selection can

become entangled with problem difficulty. This motivates selection targeted at difficult-tail cases.

## 5.5. Summary

These analyses show that random sampling is not merely a weak baseline but a stable default for multi-solution code SFT. A plausible explanation is that modern CodeLLMs already acquire broad programming knowledge during pre-training, so SFT mainly *activates* and *aligns* latent algorithmic patterns. Random sampling may preserve representative patterns from accepted solution pools, whereas diversity-maximizing objectives can overemphasize atypical implementations. This suggests that selection mainly balances representative coverage against targeted tail coverage. Accordingly, stronger objectives should be viewed as tools for shifting coverage, not as uniformly better replacements for random sampling. At the same time, the difficulty- and constraint-focused results clarify the boundary: complex selection is most useful when evaluation targets hard-tail or constraint-focused coverage rather than broad average-case performance.

## 6. Related Work

**Code Generation with LLMs.** LLMs (Brown et al., 2020; Achiam et al., 2023) have demonstrated strong capabilities in code generation (Jiang et al., 2026; Roziere et al., 2023). Chen et al. (2021) introduced Codex and the HumanEval benchmark, establishing Pass@$k$ as the standard evaluation metric. Subsequent work has scaled both model size and training data, yielding systems such as CodeLlama (Roziere et al., 2023), StarCoder (Li et al., 2023; Lozhkov et al., 2024), DeepSeek-Coder (Guo et al., 2024), and Qwen2.5-C (Yang et al., 2024). SFT has emerged as a critical technique for enhancing code generation (Albalak et al., 2024). WizardCoder (Luo et al., 2024) employs Evol-Instruct to iteratively enhance instruction complexity, while Magicoder (Wei et al., 2024b) introduces OSS-Instruct for synthesizing training data from open-source code. Self-CodeAlign (Wei et al., 2024a) uses base models to infer aligned data, and recently Liu et al. (2025) demonstrated that generating code before reasoning is better. ARBench (Liu et al., 2026a) further evaluates whether CodeLLMs can implement algorithms beyond high-level API calls. Recent work questions whether pass-rate rewards provide reliable optimization signals for code generation (Li et al., 2026).

**Data Selection and Coreset Methods.** Data selection aims to identify informative subsets for efficient training (Mirzasoleiman et al., 2020), with roots in active learning (Settles, 2009) and coreset construction (Feldman, 2020; Har-Peled & Mazumdar, 2004). K-Center Greedy (Sener & Savarese, 2018) selects samples that maxi-

mize the minimum distance to the selected set, providing a 2-approximation (Gonzalez, 1985). Facility location methods (Wei et al., 2015) provide submodular optimization frameworks for diverse subset selection. These diversity-based methods have proven effective in image classification (Sener & Savarese, 2018) and natural language understanding (Yuan et al., 2020). Du et al. (2025) propose AST Coverage, a greedy maximization approach that maximizes structural pattern coverage. Kernel Herding (Chen et al., 2010) minimizes maximum mean discrepancy between the selected subset and the full distribution, while IFD Ranking (Li et al., 2024) selects samples based on instruction-following difficulty scores. Liu et al. (2026b) rank generated code solutions by combining generated tests with static-dynamic signals. Sun et al. (2026) instead use leave-one-out consistency criteria to score tests for candidate selection.

**Data Selection for LLM Training.** Recent work has explored data selection for LLM pretraining and fine-tuning (Albalak et al., 2024). DoReMi (Xie et al., 2023a) learns domain weights for pretraining data mixture, while DSIR (Xie et al., 2023b) selects pretraining data based on importance resampling. Deduplication (Lee et al., 2022; Abbas et al., 2023) removes near-duplicate examples to improve training efficiency. LIMA (Zhou et al., 2023) demonstrates that carefully curated examples can achieve strong performance with small data. These methods are complementary to our setting, where the selection unit is a verified correct solution for each programming problem.

## 7. Conclusion

This work challenges the conventional wisdom that complex selection strategies necessarily improve data efficiency in supervised fine-tuning. Through systematic evaluation across four base models and validation on a larger size, we demonstrate that simple random sampling achieves competitive performance across all models, exhibiting better cross-model stability than complex selection methods while requiring no computational overhead. Our ablation studies also reveal that the diversity optimization method has a greater impact than the metric function within each family, and that problems with more diverse solution pools are inherently more complex, which explains why diversity-maximizing strategies can inadvertently bias training toward complex examples. These findings have direct practical implications: when training code-generation models on datasets with multiple verified solutions per problem, random sampling can serve as a strong baseline. Selectors target tails. Our study focuses on verifiable multi-solution Python algorithmic problems; extending the analysis to other languages, domains, single-solution datasets, or unverified code corpora remains future work.

## Acknowledgements

This work was supported by the Major Program (JD) of Hubei Province (2023BAA024), the Fundamental and Inter-disciplinary Disciplines Breakthrough Plan of the Ministry of Education of China (JYB2025XDXM118), the "111 Center" (B26023), and the National Natural Science Foundation of China (625B2089).

## Impact Statement

This paper presents work aimed at advancing the field of Machine Learning. Our findings suggest that, in verifiable multi-solution code-generation training pools, simple random sampling can serve as a competitive low-cost default for SFT. When more expensive selection algorithms are used, their cost should be justified by a targeted objective, such as hard-tail or constraint-focused coverage, rather than by an assumed efficiency gain. This may reduce unnecessary data-curation cost and associated energy use in relevant LLM training pipelines. We identify no negative societal impacts or dual-use concerns specific to this work, as the methods are general-purpose data-selection procedures and do not introduce new capabilities or risks.

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

# A. Experimental Setup

This section provides complete details for reproducing our experiments.

## A.1. Training Configuration

Table 13 lists all hyperparameters used in our experiments. We adopt standard practices for instruction fine-tuning without hyperparameter search; all values remain fixed across experiments to ensure fair comparison between selection methods.

*Table 13.* Training hyperparameters for all experiments.

| Hyperparameter | Value |
|---|---|
| Optimizer | AdamW |
| Learning rate | $1 \times 10^{-5}$ |
| Weight decay | 0.01 |
| $\beta_1, \beta_2$ | $0.9, 0.999$ |
| Warmup ratio | 10% of total steps |
| LR scheduler | Cosine decay |
| Batch size | 128 |
| Epochs | 3 |
| Max sequence length | 2,048 tokens |
| Precision | bfloat16 |
| Gradient clipping | 1.0 |

## A.2. Selection Method Implementation

**Embedding Generation.** For embedding-based methods (Cosine K-Center, K-Means, Kernel Herding), we use Code-BERT (Feng et al., 2020) to generate 768-dimensional embeddings with mean pooling over final hidden states.

**Distance and Similarity Metrics.** K-Center methods use the following distance functions:

- **Cosine**: $d(\mathbf{u}, \mathbf{v}) = 1 - \frac{\mathbf{u} \cdot \mathbf{v}}{\|\mathbf{u}\| \|\mathbf{v}\|}$ on CodeBERT embeddings
- **Levenshtein**: Token-level edit distance using Python's `tokenize` module
- **Jaccard**: $d(A, B) = 1 - \frac{|A \cap B|}{|A \cup B|}$ on token sets
- **Syntax**: Jaccard distance on AST fingerprints extracted via tree-sitter

**Computational Cost.** Table 14 reports wall-clock time for each selection method. Random sampling incurs negligible overhead, while Levenshtein K-Center requires the most computation due to pairwise distance calculations.

*Table 14.* Computational cost of selection methods (2,641 problems, 154K solutions).

| Method | Preprocess | Selection |
|---|---|---|
| Random | 0 min | <1 min |
| Cosine K-Center | 30 min (embed) | 18 min |
| Levenshtein K-Center | 0 min | 20 min |
| K-Means | 30 min (embed) | <1 min |
| AST Coverage | 12 min (parse) | 30 min |
| Kernel Herding | 30 min (embed) | 15 min |
| IFD Ranking | 60 min (loss) | <1 min |
| Facility Location | 30 min (embed) | 20 min |

## A.3. Reproducibility

**Code and Data.** All code for data selection, model training, and evaluation will be released under the MIT license. The LeetCode dataset derives from publicly available submissions (Xia et al., 2025; Morales, 2024).

**Random Seeds.** We use seed 347 for main experiments (Table 2) and seeds $\{347, 348, 349, 350, 351\}$ for statistical significance tests.

**Hardware.** All experiments run on NVIDIA A100 GPUs (80GB) with CUDA 12.1 and PyTorch 2.1. Each 7B model run completes in ∼64 GPU-hours; 32B experiments use 8 GPUs with tensor parallelism (∼256 GPU-hours).

## B. Dataset

### B.1. Statistics and Preprocessing

Our training dataset is derived from LeetCode (Xia et al., 2025; Morales, 2024), comprising 2,641 problems across multiple difficulty levels and algorithmic categories. Table 15 provides detailed statistics.

*Table 15.* Dataset statistics. Problems are categorized by difficulty level and algorithmic topic. Each problem may have multiple topic tags, so topic percentages sum to more than 100%.

| Category | Count | Percentage |
|---|---|---|
| *By Difficulty* | | |
| Easy | 638 | 24.2% |
| Medium | 1,397 | 52.9% |
| Hard | 606 | 22.9% |
| *By Topic (Top Categories)* | | |
| Array / Hash Table | 1,783 | 67.5% |
| Dynamic Programming | 511 | 19.3% |
| String | 675 | 25.6% |
| Math | 482 | 18.3% |
| Sorting / Greedy | 637 | 24.1% |
| Graph / Tree | 404 | 15.3% |
| **Total** | 2,641 | 100% |

We apply MinHash LSH deduplication with a Jaccard threshold of 0.85 to remove near-duplicates and cap each problem at 100 solutions. Table 16 shows the distribution before and after preprocessing: the raw dataset has a median of 63 and mean of 191 solutions per problem, while after processing the median becomes 60 and the mean drops to 59 due to the cap.

*Table 16.* Distribution of solutions per problem before and after deduplication. Raw counts are from the original LeetCode submissions; after MinHash LSH deduplication each problem retains at most 100 solutions.

| Solutions / Problem | Before Dedup | | After Dedup | |
|---|---|---|---|---|
| | Count | % | Count | % |
| 1 | 212 | 8.0% | 213 | 8.1% |
| 2–10 | 237 | 9.0% | 240 | 9.1% |
| 11–50 | 723 | 27.4% | 740 | 28.0% |
| 51–100 | 461 | 17.5% | 1,448 | 54.8% |
| 101–500 | 776 | 29.4% | 0 | 0.0% |
| 501–1,000 | 130 | 4.9% | 0 | 0.0% |
| >1,000 | 102 | 3.9% | 0 | 0.0% |
| **Total problems** | 2,641 | | 2,641 | |
| **Total solutions** | 503,268 | | 154,714 | |
| **Mean** | 191 | | 59 | |
| **Median** | 63 | | 60 | |

### B.2. Data Characteristics

Figure 3 shows solution length distributions in lines of code. Before deduplication, solutions have a median of 15 lines and a mean of 17 lines; after deduplication, the median decreases to 13 lines and the mean to 15 lines, while the overall distribution shape remains similar.

Figure 4 shows how solution counts per problem change through preprocessing. The raw dataset (left, log scale) exhibits extreme variation; after deduplication and capping (right), the distribution becomes more balanced.

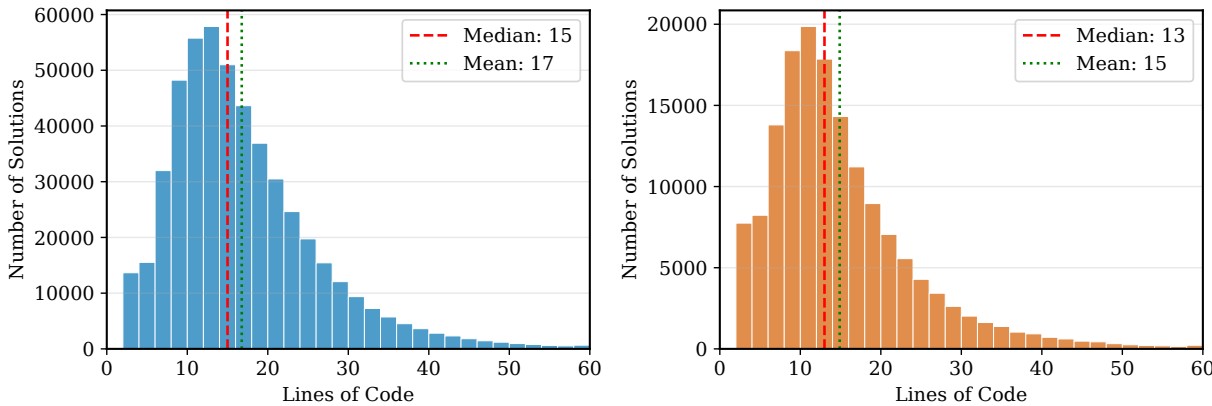

*Figure 3.* Distribution of solution lengths (lines of code). Left: before deduplication. Right: after deduplication.

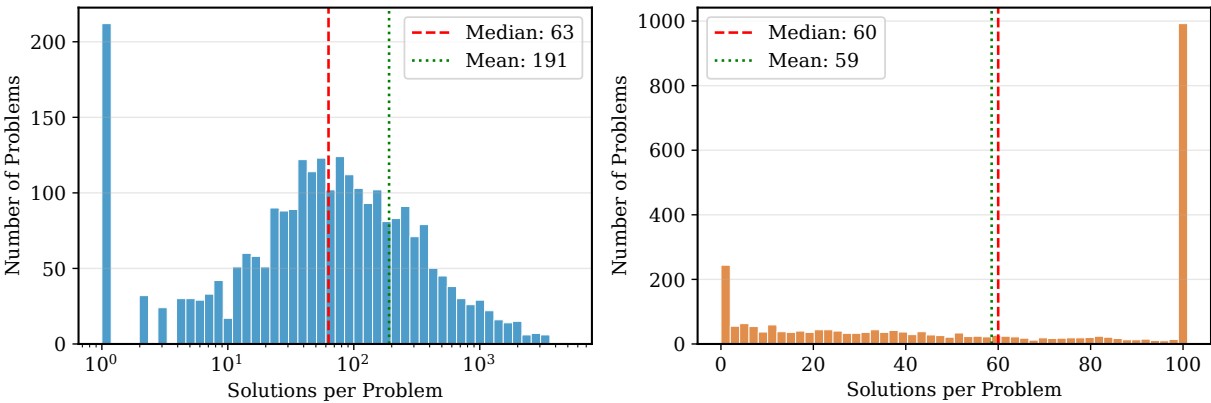

*Figure 4.* Distribution of solutions per problem. Left: before deduplication, with median 63 and mean 191. Right: after deduplication, with median 60 and mean 59, capped at 100.

*Table 17.* Diversity metrics and Pass@k by problem category. Categories are determined by primary algorithmic technique required. Dynamic Programming and Tree & Graph problems show higher diversity and lower Pass@k, while Array & String problems are relatively easier with lower diversity.

| Category | $n$ | Levenshtein | Jaccard | Syntax | Pass@1 |
|---|---|---|---|---|---|
| Array/String | 33 | 122.1 | 0.43 | 0.72 | 16.4% |
| Math/Bit | 35 | 145.9 | 0.49 | 0.74 | 15.2% |
| Data Structures | 27 | 152.4 | 0.47 | 0.73 | 3.3% |
| Tree/Graph | 24 | 258.7 | 0.54 | 0.77 | 2.1% |
| Dynamic Programming | 52 | 196.4 | 0.48 | 0.76 | 0.2% |
| Greedy/Backtrack | 25 | 192.6 | 0.50 | 0.76 | 0.0% |

# C. Additional Analysis

This section presents empirical and theoretical analyses that support the main findings.

## C.1. Diversity–Difficulty Correlation

We hypothesize that problems with more diverse solution pools are inherently harder, explaining why diversity-maximizing methods underperform. Table 17 groups test set metrics by algorithmic category. Array & String problems tend to have lower diversity and the highest Pass@1. In contrast, Dynamic Programming, Tree & Graph, and Greedy & Backtracking problems exhibit higher diversity and substantially lower Pass@1, reflecting their algorithmic complexity. These patterns suggest that diversity-based selection may inadvertently favor harder problems, in which models struggle to generalize.

### C.2. Theoretical Perspectives

We provide intuitions (not formal proofs) for why random sampling performs well.

**Sufficient Coverage.** Consider solutions partitioned into $C$ clusters, each representing an algorithmic approach. The coupon collector problem (Motwani & Raghavan, 1995) shows that $O(C \log C)$ random samples cover all clusters with high probability. Empirically, K-Means elbow analysis yields $C$ between 3 and 5 for typical problems. With $k = 11$ samples and $C \leq 5$, the probability of missing any cluster is <5%.

**Bias–Variance Tradeoff.** Diversity maximization introduces selection bias toward boundary points, reducing coverage variance but increasing bias by under-representing typical solutions. When solution pools exhibit sufficient natural variation, this bias outweighs the benefits of variance reduction.

**Distribution Alignment.** Random sampling ensures $P_{\text{train}} \approx P_{\text{pool}}$, preserving the original distribution. Diversity-based selection creates $P_{\text{train}} \neq P_{\text{pool}}$, introducing a distribution shift (Quiñonero-Candela et al., 2009) that harms generalization when the original distribution reflects meaningful patterns.

## D. Limitations

**Scope.** Our study focuses on Python code generation for algorithmic problems. Generalization to other languages, domains, or task formats remains untested.

**Dataset Assumptions.** Our conclusions apply to settings with *multiple verified correct solutions* per problem. The advantage of random sampling may not hold for single-solution datasets or unverified code.

**Model Scale.** While we validate on models up to 32B parameters, extrapolation to larger scales (70B+) remains untested.

