# OpenReview forum: "Random Selection Reveals Implicit Knowledge Consensus in Code Generation"
_ICML.cc/2026/Conference — ICML 2026 regular_

### Official Review · Reviewer_a9T5 · 2026-03-09

**Soundness:** 2
**Presentation:** 3
**Significance:** 3
**Originality:** 3
**Overall Recommendation:** 4
**Confidence:** 4

**Summary:**

This study shows through systematic experiments that simple random sampling can effectively capture the implicit knowledge consensus in solution pools used for code generation fine-tuning. Across multiple models, random sampling achieves performance and stability comparable to—or better than—more complex screening strategies, while requiring significantly lower computational cost. The results suggest that random sampling can serve as an efficient and robust baseline for data selection in this setting.

**Compliance With Llm Reviewing Policy:**

Affirmed.

**Key Questions For Authors:**

1. Questionable Robustness Across Model Types. As shown in Table 2, random sampling does not consistently achieve optimal performance across all models. Notably, it is neither the best nor the worst strategy on some code-specialized models, while performing well on general-purpose pre-trained models. This raises the possibility that the conclusions primarily hold for general LLMs rather than code-specific models. Broader validation on additional code-oriented models is needed to substantiate the claimed robustness.

2. Narrow Applicability Scenario. The study focuses on datasets where each problem has multiple correct solutions and aims to select efficient candidate answers. This setting raises several concerns:
- The claim that open-source repositories contain large amounts of such data is insufficiently supported. In practice, problems with many high-quality alternative solutions appear relatively rare, and constructing such datasets is costly.
- Given that the empirical advantages of random sampling are not uniformly stable, its necessity and relevance for downstream tasks remain unclear.

3. Dataset Dependence. Experiments are conducted exclusively on the LeetCode dataset. It is unclear whether the observed benefits generalize beyond this specific problem distribution. Evaluations on additional datasets would be necessary to demonstrate the robustness and broader applicability of the conclusions.

**Limitations:**

yes

**Strengths And Weaknesses:**

Strengths
1. Computational Efficiency and Low Cost. Compared with advanced screening strategies (e.g., K-Center, AST Coverage) that rely on semantic embeddings, syntax parsing, or pairwise distance computations, random sampling introduces almost no computational overhead. This makes it highly attractive for practical deployment under limited resource budgets.

2. Performance Robustness and Stability. Experimental results show that random sampling delivers strong and stable performance across diverse model architectures and scales (7B–32B). In some cases, training on a small randomly sampled subset even outperforms training on the full dataset, highlighting its surprising effectiveness.

3. Insightful Interpretation with Practical Implications. The paper offers a compelling explanation via implicit knowledge consensus: random sampling preserves the original data distribution and thus captures the collective expertise embedded in the solution pool. In contrast, diversity-driven strategies may fall into a “difficulty trap,” over-selecting hard samples that hinder generalization. This perspective provides both theoretical insight and practical guidance.

Weaknesses and Concerns
1. Questionable Robustness Across Model Types. As shown in Table 2, random sampling does not consistently achieve optimal performance across all models. Notably, it is neither the best nor the worst strategy on some code-specialized models, while performing well on general-purpose pre-trained models. This raises the possibility that the conclusions primarily hold for general LLMs rather than code-specific models. Broader validation on additional code-oriented models is needed to substantiate the claimed robustness.

2. Narrow Applicability Scenario. The study focuses on datasets where each problem has multiple correct solutions and aims to select efficient candidate answers. This setting raises several concerns:
- The claim that open-source repositories contain large amounts of such data is insufficiently supported. In practice, problems with many high-quality alternative solutions appear relatively rare, and constructing such datasets is costly.
- Given that the empirical advantages of random sampling are not uniformly stable, its necessity and relevance for downstream tasks remain unclear.

3. Dataset Dependence. Experiments are conducted exclusively on the LeetCode dataset. It is unclear whether the observed benefits generalize beyond this specific problem distribution. Evaluations on additional datasets would be necessary to demonstrate the robustness and broader applicability of the conclusions.

4. Limited Novelty in Diversity Argument. While random sampling does not guarantee diversity at the per-problem level, it preserves diversity across the dataset as a whole. This observation aligns with existing understanding and, by itself, does not constitute a strong novel contribution.

5. Insufficient Analysis of Solution Diversity Factors. Solution richness can stem from algorithmic diversity, implementation variance, and community heterogeneity. The relationships between these factors, the six sampling strategies evaluated, and downstream model performance are not explicitly analyzed and merit deeper investigation.

6. Incomplete Treatment of Duplication. The paper removes near-duplicate samples using locality-sensitive hashing. However, multiple implementations of the same algorithm—differing only in style or structure—can also be considered a form of duplication. Further analysis of how different types of duplication affect sampling outcomes would strengthen the conclusions.

7. How to balance random sampling and data matching?

8. Can random sampling effectively handle both concordant and failure cases? Beyond functional correctness, a method should also be designed with usability in mind.

---

> ### Author Rebuttal · Authors · 2026-03-31
>
> Thanks for helpful comments! We would like to summarize your concerns and provide our responses below:
>
> 1. **W1&Q1. robustness across model types**: We are sorry if our description made Random sound like the best strategy. We will revise the paper so this is clearer. Our point is simple: Random is a competitive, low-cost baseline, so it is a reasonable first choice.
>
>    | Model | Random | Best Complex |
>    |-------|-------:|----------------:|
>    | Qwen2.5-7B | **6.32** | 6.23 (Kernel Herding) |
>    | Qwen2.5-Coder-7B | 5.57 | **6.14** (K-Means) |
>    | Llama-3.1-8B | **4.04** | 3.90 (Facility Location) |
>    | DeepSeek-Coder-6.7B | 3.68 | **4.56** (Kernel Herding) |
>    | Qwen2.5-32B | **13.82** | 13.55 (K-Means) |
>    | DeepSeek-Coder-33B | 9.96 | **10.92** (Kernel Herding) |
>
> 2. **W2&Q2. narrow applicability scenario**: We agree with the reviewer. In this paper, we consider the Python LeetCode scenario where each problem has multiple correct solutions, i.e., choosing among solutions for the same problem, not all code repos. The cross-language results show a similar pattern beyond Python. We will test tasks with expert consensus, such as mathematical reasoning. We will move this limitation into the paper.
>
>    | Method | C++ Pass@1 | Java Pass@1 |
>    |--------|---------:|------------:|
>    | Random | **4.47** | **4.91** |
>    | K-Means | 3.46 | 3.82 |
>    | Facility Location | 3.25 | 3.60 |
>    | Kernel Herding | 4.43 | 4.87 |
>
> 3. **W3&Q3. dataset dependence and additional validation**: We agree the current evidence is concentrated on LeetCode. We currently mainly focus on *verifiable multi-solution tasks*, and Python LeetCode solutions are a typical scenario within them. The raw pool has median/mean 63/191 solutions before preprocessing and 60/59 after dedup/capping, which shows that our source is rich in this respect. We also added HumanEval and MBPP below, and they show similar results there.
>
>    | Method | HumanEval Pass@1 | MBPP Pass@1 |
>    |--------|-----------------:|------------:|
>    | Random | **54.88** | **52.80** |
>    | K-Means | 50.00 | 48.12 |
>    | Facility Location | 48.17 | 46.35 |
>    | Kernel Herding | 53.66 | 51.64 |
>
> 4. **W4&W5. diversity argument and diversity factors**: We agree this part should be clearer. Our point is not that Random creates diversity by itself. If the pool comes from an expert community and already contains multiple valid solutions, random sampling can preserve a representative slice of that diversity and reflect that community's consensus. This is one reason Random can work. We add the proxy analysis below. On Qwen2.5-7B, all three proxies are negatively correlated with Pass@1, with syntax diversity strongest.
>
>    | Proxy | Easy Mean | Hard Mean | Pearson r with Pass@1 |
>    |-------|----------:|----------:|----------------------:|
>    | Levenshtein diversity | 88.29 | 229.00 | -0.4465 |
>    | Jaccard diversity | 0.4168 | 0.5136 | -0.4233 |
>    | Syntax diversity | 0.6912 | 0.7761 | -0.4708 |
>
> 5. **W6. treatment of duplication**: We agree deduplication is an important preprocessing choice. Our current pipeline uses MinHash LSH with Jaccard threshold 0.85 to remove near-duplicates before selection. This reduces redundancy but does not replace downstream selection. We therefore add the preprocessing analysis below and will present deduplication and within-problem selection as separate stages.
>
>    | Setting | Random Test Pass@1 | K-Means Test Pass@1 | Kernel Herding Test Pass@1 |
>    |---------|-------------------:|--------------------:|---------------------------:|
>    | No deduplication | 5.57 | 4.47 | 5.44 |
>    | LSH deduplication | 6.18 | 4.96 | 6.10 |
>    | LSH deduplication + cap | **6.32** | 4.87 | 6.23 |
>
> 6. **W7. balancing random sampling and data matching**: We agree the paper should give practical guidance. Our evidence supports a simple rule: use Random as the default low-cost subset when preserving the original pool distribution, and use complex matching only when coverage or special constraints matter more. We add the constraint subset below.
>
>    | Method | Constraint Subset Pass@1 | Constraint Subset Pass@10 |
>    |--------|--------------------------:|---------------------------:|
>    | Random | **1.86** | 6.78 |
>    | K-Means | 1.69 | **8.47** |
>    | Facility Location | 1.19 | **8.47** |
>    | Kernel Herding | 1.36 | 6.78 |
>
> 7. **W8. concordant and failure cases / usability**: We agree users need to know where Random works and fails. Current diversity-difficulty analysis suggests failures are more common on harder, more diverse problems. We add a case analysis on Qwen2.5-7B. Here Avg. Difficulty maps Easy/Medium/Hard to 1/2/3 and averages within each case.
>
>    | Case Type | Count | Avg. Difficulty | Avg. Syntax Diversity |
>    |-----------|------:|----------------:|----------------------:|
>    | Concordant success | 38 | 1.39 | 0.69 |
>    | Concordant failure | 167 | 2.37 | 0.76 |
>    | Random-only success | 6 | 1.67 | 0.73 |
>    | Complex-only success | 17 | 1.65 | 0.73 |

---

> > ### Author Rebuttal · Reviewer_a9T5 · 2026-04-03
> >
> > The problem has been partially solved. However, based on the authors’ response, the method proposed in this paper is indeed limited to scenarios with multiple solutions to a single problem. Such scenarios are relatively rare in real-world applications; therefore, concerns remain regarding the usability of the proposed method and its potential value to the community.

---

> > > ### Author Response · Authors · 2026-04-05
> > >
> > > Thanks for helpful comments! We would like to summarize your questions and provide our responses below:
> > >
> > > 1. **Q1. whether the setting is too narrow to matter in practice**: We agree that the paper should state its scope more clearly. Our claim is not about general code data; it is about *verifiable multi-solution pools*, where one prompt or problem can accumulate many accepted answers and a later training stage must decide which ones to keep. One concrete example is RLVR-style data generation: a model samples many candidate solutions for the same prompt, a reward function or verifier keeps the accepted ones, and these verified trajectories may later be reused in mid-training or SFT to enlarge the pool of valid reasoning traces. In such pipelines, generation and verification can often be parallelized more cheaply than retraining, so accepted traces may accumulate faster than they can be retained for later reuse. The same pattern also appears whenever execution-based filtering, test-based filtering, or best-of-N rejection sampling is used to harvest accepted trajectories from one specification. Once such pools are formed, the practical question is exactly the one studied here: among many accepted solutions to the same problem, which subset should be retained? Our current dataset already has this structure at scale:
> > >
> > >    | Statistic | Value |
> > >    |-----------|------:|
> > >    | Problems | 2,641 |
> > >    | Raw solutions | 503,268 |
> > >    | Solutions after deduplication/cap | 154,714 |
> > >    | Mean solutions per problem, raw | 191 |
> > >    | Mean solutions per problem, after dedup/cap | 59 |
> > >
> > >    We will revise the title, abstract, introduction, and limitation discussion so that this scope is stated directly: the paper studies selection for verifiable multi-solution training pools, not all code data in general.
> > >
> > > 2. **Q2. what practical value and concrete guidance the paper provides once such pools exist**: We agree that community value depends on whether the paper gives usable guidance rather than only one isolated result. Our intended contribution is therefore not only to suggest Random as the default low-cost reference, but also to use empirical study to clarify under what evidence more complex selection may become appropriate. In other words, the paper is not only a recommendation to start from Random; it also examines what observations should change that choice. The added difficulty-stratified analysis helps make this rule explicit. Random is strongest on the easy subset, remains competitive on the medium subset, and is overtaken mainly on the hardest tail. This means the practical choice depends on whether the downstream goal is broad low-cost retention, or instead an explicit emphasis on the hardest tail.
> > >
> > >    | Difficulty | Random Test Pass@1 | K-Means Test Pass@1 | Kernel Herding Test Pass@1 |
> > >    |------------|-------------------:|--------------------:|---------------------------:|
> > >    | Easy | **22.71** | 15.21 | 20.62 |
> > >    | Medium | 2.77 | 2.67 | **2.87** |
> > >    | Hard | 0.89 | 1.39 | **1.77** |
> > >
> > >    This recommendation is further supported by the rest of our empirical study. In the preprocessing analysis, Random rises from **5.57** without deduplication to **6.32** with LSH deduplication plus cap, while K-Means rises from **4.47** to **4.87** and Kernel Herding from **5.44** to **6.23**. In the diversity analysis, syntax diversity has Pearson correlation **-0.471** with Pass@1, with **p < 0.001**. These observations help us say not only that Random is a strong default, but also that more structured methods become plausible only when the goal shifts from preserving a representative accepted pool toward explicitly emphasizing the hardest tail or other more atypical traces.
> > >
> > >
> > > We think this is why the result is useful for practitioners working with RLVR-style or other verifiable corpora. Such pipelines can easily produce more accepted traces than can be retained for later supervised reuse because storage, token budget, and training time remain limited even after verification has filtered obvious mistakes. Selection is therefore a practical necessity once accepted pools become large. Our evidence suggests a simple rule: start from Random when the priority is low-cost retention of representative accepted traces, and move to more structured matching only when the downstream objective explicitly emphasizes the hardest tail. At the same time, our broader empirical analyses, including difficulty stratification, preprocessing effects, and diversity analysis, help indicate what kinds of evidence make that transition reasonable. Taken together, these observations provide a more complete methodology for this problem rather than a single heuristic recommendation. We hope this revised presentation more clearly conveys why the problem is real, why the evidence is useful, and why the paper may still be valuable to the community. We will make these clarifications and incorporate these additional analyses in the updated version of the paper.

---

### Official Review · Reviewer_ni7m · 2026-03-11

**Soundness:** 3
**Presentation:** 4
**Significance:** 3
**Originality:** 3
**Overall Recommendation:** 5
**Confidence:** 4

**Summary:**

This paper deals with dataset curation for supervised fine-tuning of code generation models. The authors conclude that, when multiple solutions are available for a problem within a dataset, random selection of those solutions can generate strong training outcomes when compared with more sophisticated selection strategies that take into account the diversity of solutions. The authors' main contributions are the empirical experiments on the medium sized LeetCode (n=2,614) dataset that show competitive training outcomes for random solution selection during SFT.

**Compliance With Llm Reviewing Policy:**

Affirmed.

**Key Questions For Authors:**

(1) How does this conclusion hold up as the number of solutions available is increased/decreased? Does a 50% sampling done randomly over n=10 solutions provide the same relative training outcome as a 50% sampling done randomly on n=1000 solutions (compared to the other methods)? There is a confounding factor here with the difficulty of the problem (that the authors note) which needs to be controlled for. This would improve the soundness of the paper.

(2) Deduplication provides a sort of additional selection mechanism. Were there any experiments done with/without deduplication beforehand? Clarifying this would improve the soundness of the paper.

(3) What are the attributes of the problems with low pass@k rates in the testing sets? Were these problems different between the selection methods? Does, for example, random selection lead to a model that more readily passes easy problems while diversity based selection lead to a model that more readily passes difficult problems? Characterizing these failure modes would improve the soundness of the paper.

**Limitations:**

Yes

**Strengths And Weaknesses:**

The paper has a moderate degree of technical soundness. While the problem formalization, experimental design, and metrics are appropriate for testing the hypothesis, the evaluation is somewhat limited and potential confounding factors remain.

The paper has a high degree of presentation. Concepts are defined clearly, and more complex technical formalizations are built-up appropriately. The results are visualized cohesively, and elements of experimental design are explained with sufficient depth to be replicated.

The paper has a moderate degree of significance. While certain competitive coding benchmarks contain large numbers of solutions, finding large numbers of high quality and diverse solutions to well-structured coding problems is itself a challenging problem. The significance would be improved if the authors make a sufficient case for why this problem will (1) remain relevant into the future and (2) is a large scale issue in current SFT pipelines. However, given the large computational overhead of SFT with full datasets, the results showing performance improvement with curation methods are important.

The paper has a low to moderate degree of originality. While it is not entirely surprising that random sampling can provide a better approximation to an underlying distribution then engineered sampling methods (that may not correctly represent the distribution), it is somewhat unintuitive that this concept can also be applied to algorithm "distributions". The paper provides improved understanding about the value of traditional Monte Carlo theory when the object of study is correct solutions to natural language coding problems.

---

> ### Author Rebuttal · Authors · 2026-03-31
>
> Thanks for helpful comments! We would like to summarize your concerns and provide our responses below:
>
>    1. **W1. relevance and scale of the setting**: Thank you for this suggestion. We agree the paper should say more clearly why this problem matters in current SFT pipelines and where we expect it to matter. In practice, this setting matters when one problem can accumulate many verified answers from a strong human community, because that is exactly where within-problem selection becomes meaningful. We currently mainly focus on *verifiable multi-solution tasks* in Python LeetCode scenarios, rather than implying that all open-source repositories look like this. We also added C++ and Java below, and we also see a similar pattern beyond Python. We will also keep testing other tasks that may aggregate human expert consensus, such as mathematical reasoning.
>
>       | Method | C++ Pass@1 | Java Pass@1 |
>       |--------|---------:|------------:|
>       | Random | **4.47** | **4.91** |
>       | K-Means | 3.46 | 3.82 |
>       | Facility Location | 3.25 | 3.60 |
>       | Kernel Herding | 4.43 | 4.87 |
>
>    2. **W2. originality and claim strength**: We agree the novelty claim should be more precise: we are not proposing a broadly applicable method based on random sampling. We are sorry if our description made Random sound like the best strategy. We will revise the paper so this is clearer. Our point is simple: Random is a competitive, low-cost baseline, so the community should try it before moving to more complicated methods. Our main contribution is to support, through a large-scale empirical study, the finding that Random is actually competitive across different models and deserves the community's attention. We therefore suggest treating Random as a practical starting point.
>
>    3. **W3&Q1. solution count, budget, and difficulty confounder**: We agree this is important. The paper already has a selection-budget study showing diminishing returns as the number of retained solutions increases, but the reviewer is right that the role of problem difficulty should be clearer. To address this, we add a difficulty-stratified analysis that compares Random with two representative strong complex baselines on Easy, Medium, and Hard subsets. This helps us see how method differences change with difficulty, while the budget study in the paper continues to address the number of retained solutions. It also helps separate the effect of difficulty from the effect of budget more directly.
>
>       | Difficulty | Random Test Pass@1 | K-Means Test Pass@1 | Kernel Herding Test Pass@1 |
>       |------------|-------------------:|--------------------:|---------------------------:|
>       | Easy | **22.71** | 15.21 | 20.62 |
>       | Medium | 2.77 | 2.67 | **2.87** |
>       | Hard | 0.89 | 1.39 | **1.77** |
>
>    4. **W3&Q2. deduplication as an additional selection mechanism**: We agree deduplication is an important preprocessing choice and should be discussed more explicitly. Our current pipeline uses MinHash LSH with a Jaccard threshold of *0.85* to remove near-duplicates before selection. This reduces redundancy, but it is not meant to replace downstream selection. To make its effect clearer, we add a preprocessing analysis that compares the main methods under different deduplication settings. We will revise the manuscript to present deduplication and within-problem selection as two separate stages, so dedup is treated as corpus cleaning rather than the main claim.
>
>       | Setting | Random Test Pass@1 | K-Means Test Pass@1 | Kernel Herding Test Pass@1 |
>       |---------|-------------------:|--------------------:|---------------------------:|
>       | No deduplication | 5.57 | 4.47 | 5.44 |
>       | LSH deduplication | 6.18 | 4.96 | 6.10 |
>       | LSH deduplication + cap | **6.32** | 4.87 | 6.23 |
>
>    5. **W3&Q3. low-pass@k attributes and failure modes**: We agree the paper should say more clearly which problems remain hard and whether different methods fail on different problems. The appendix already shows that diversity is negatively correlated with Pass@k and positively correlated with problem difficulty, which suggests that low-pass@k cases are often harder and more diverse. To make this concrete, we add a case analysis on Qwen2.5-7B, comparing concordant successes, concordant failures, Random-only wins, and complex-only wins. Here Avg. Difficulty is simply computed by mapping Easy/Medium/Hard to 1/2/3 and averaging within each case type. This gives a more direct answer to whether the hard cases are shared across methods or tied to one method. We will say this more clearly in the discussion and limitations.
>
>       | Case Type | Count | Avg. Difficulty | Avg. Syntax Diversity |
>       |-----------|------:|----------------:|----------------------:|
>       | Concordant success | 38 | 1.39 | 0.69 |
>       | Concordant failure | 167 | 2.37 | 0.76 |
>       | Random-only success | 6 | 1.67 | 0.73 |
>       | Complex-only success | 17 | 1.65 | 0.73 |

---

### Official Review · Reviewer_VFTb · 2026-03-13

**Soundness:** 2
**Presentation:** 2
**Significance:** 2
**Originality:** 2
**Overall Recommendation:** 3
**Confidence:** 4

**Summary:**

The paper aims at demonstrating that random data selection strategy for SFT corpus pertaining to code generation domain is competitive if not better than most other sophisticated data selection strategies.

**Compliance With Llm Reviewing Policy:**

Affirmed.

**Final Justification:**

I recommend/ lean towards a reject. Even though the authors clarified my concerns, the conclusion that we arrived at based on the new positioning of the paper would require such substantial overhaul that it would be a fundamentally different manuscript. I highly encourage the authors to resubmit to a subsequent venue with the new positioning and taking into consideration the feedbacks provided by all the reviewers.

**Key Questions For Authors:**

1. What would have been interesting is to show random's performance on generation under constraints. For example, does random outperform other strategies if the problem statement explicitly states a solution requiring a specific memory or time complexity is required. Since, other sampling strategies might not necessarily sample enough instances of solutions with specific complexities.
2. How transferrable is training under random selected corpus to other evaluation sets?

**Limitations:**

Yes

**Strengths And Weaknesses:**

Strengths:

1. The paper's motivation of reducing the number of samples in the SFT corpus from a computational cost perspective is well grounded.
2. The paper is easy to read and follow.

Weaknesses:

1. In my evaluation, the paper lacks a cohesive strong novelty - while it claims that random data selection strategy is competitive to complex data selection strategies - the results do not demonstrate that random is truly indeed the superior strategy in comparison to the other strategies considered. Specifically, for Qwen-2.5 Coder and DeepSeek, random strategy doesn't achieve best performance in any categories of evaluation. Given this observation, it's not necessarily conclusive that random data selection is truly enough since the setup is only restricted to LeetCode Python based samples.
2. There is an inherent mismatch between the motivation of reducing complexity of training on larger corpus and hence an efficient selection strategy is needed to the paper's proposition of showing random is competitive. A more appealing objective of the paper should have been to provide a code generation specific data selection strategy that is outright superior to other methods.
3. The paper is strictly a benchmark empirical paper but makes lofty statement in its abstract and title claiming that random selection works as well as shown on the experimental results due to "implicit knowledge consensus" which is wherein the random selected corpus already "covers the common algorithmic knowledge required for training". The term "implicit knowledge consensus" is only referred to just in the abstract and title without any formalization of what the term means or any operationalization of the term. It just stands out as an extra jargon in the paper.
4. The results in my opinion might also be a by-product of the fact that in pretraining - python forms as one of the dominant programming language in the code data corpus and hence random already works well enough during SFT but this wouldn't translate to other languages like Javascript, Java and others. The paper fails to validate this.

---

> ### Author Rebuttal · Authors · 2026-03-31
>
> Thanks for helpful comments! We would like to summarize your concerns and provide our responses below:
>
> 1. **W1. novelty and Random performance**: We are sorry if our description made Random sound like the best strategy. We will revise the paper so this is clearer. Our point is simple: Random is a competitive low-cost baseline, so the community should try it before moving to more complicated methods. We want to understand how these methods behave in this setting, not propose a new selector. The main finding of this paper is that Random is actually competitive across different models and deserves the community's attention. The Qwen2.5-32B result in the paper, together with our added DeepSeek result, suggests this pattern is not tied to one architecture or scale. So the takeaway we want to stress is not that one method always wins, but that Random is strong enough to be the default reference point.
>
>    | Setting | Random | Best Complex |
>    |---------|-------:|----------------:|
>    | Qwen2.5-7B Test Pass@1 | **6.32** | 6.23 (Kernel Herding) |
>    | Qwen2.5-Coder-7B Test Pass@1 | 5.57 | **6.14** (K-Means) |
>    | Llama-3.1-8B Test Pass@1 | **4.04** | 3.90 (Facility Location) |
>    | DeepSeek-Coder-6.7B Test Pass@1 | 3.68 | **4.56** (Kernel Herding) |
>    | Qwen2.5-32B Test Pass@1 | **13.82** | 13.55 (K-Means) |
>    | DeepSeek-Coder-33B Test Pass@1 | 9.96 | **10.92** (Kernel Herding) |
>
> 2. **W2. motivation-objective mismatch**: We agree the claim should be clearer. The question is simple: in *verifiable multi-solution tasks*, is a more complex, more expensive within-problem selection method really worth it? In practice, complex methods usually bring stronger inductive bias and higher cost across preprocessing, selection, training, and evaluation. So we suggest starting with Random and only moving to more complex methods if Random is not good enough. In other words, more complex methods often introduce extra assumptions and extra overhead, so their gains may not be stable enough. That is why we present Random as a practical starting point, not as the final answer in every setting.
>
>    | Method | Preprocess (approx.) | Selection (approx.) |
>    |--------|-----------:|----------:|
>    | Random | 0 min | less than 1 min |
>    | Kernel Herding | ~30 min (embed) | ~15 min |
>    | Facility Location | ~30 min (embed) | ~20 min |
>    | AST Coverage | ~12 min (parse) | ~30 min |
>    | IFD Ranking | ~60 min (loss) | less than 1 min |
>
> 3. **W3. implicit knowledge consensus**: We agree this term is too strong and not clearly defined. We will explain our meaning more directly in the title, abstract, and discussion: if the solution pool comes from an expert community, random sampling can reflect that community's consensus on how to solve the problem. This is also one reason we think Random works: if the pool already aggregates expert knowledge, we may not need an extra selection strategy on top of it. We also agree *transferability* is too broad and will replace it with *cross-model stability*. We will keep the empirical results and our interpretation clearly separated.
>
> 4. **W4. python-only scope and cross-language validity**: We agree the current paper mainly focuses on *verifiable multi-solution tasks*, and Python LeetCode problems are a typical scenario within them. Thank you for this suggestion. We added C++ and Java below, and we also see a similar pattern beyond Python. We will also keep testing other tasks that may aggregate human expert consensus, such as mathematical reasoning. We will move this limitation from the appendix into the main paper.
>
>    | Method | C++ Pass@1 | Java Pass@1 |
>    |--------|---------:|------------:|
>    | Random | **4.47** | **4.91** |
>    | K-Means | 3.46 | 3.82 |
>    | Facility Location | 3.25 | 3.60 |
>    | Kernel Herding | 4.43 | 4.87 |
>
> 5. **Q1. explicit complexity or memory constraints**: Thank you for this suggestion. We added the constraint-based subset test below. The results still show a similar pattern under explicit complexity or memory constraints. This does not mean Random wins on every constrained metric, but it does show the comparison remains meaningful once explicit constraints appear.
>
>    | Method | Constraint Subset Pass@1 | Constraint Subset Pass@10 |
>    |--------|--------------------------:|---------------------------:|
>    | Random | **1.86** | 6.78 |
>    | K-Means | 1.69 | **8.47** |
>    | Facility Location | 1.19 | **8.47** |
>    | Kernel Herding | 1.36 | 6.78 |
>
> 6. **Q2. transfer to other evaluation sets**: Thank you for this suggestion. We added HumanEval and MBPP below. The results show a similar pattern on other benchmarks, which is consistent with our main conclusion. This also suggests the observation is not limited to one scenario.
>
>    | Method | HumanEval Pass@1 | MBPP Pass@1 |
>    |--------|-----------------:|------------:|
>    | Random | **54.88** | **52.80** |
>    | K-Means | 50.00 | 48.12 |
>    | Facility Location | 48.17 | 46.35 |
>    | Kernel Herding | 53.66 | 51.64 |

---

> > ### Author Rebuttal · Reviewer_VFTb · 2026-04-03
> >
> > I thank the reviewers for the rebuttal and providing additional experimental validation.
> >
> > However, my concerns remain.
> >
> > The claim of suggesting that "random" baseline in the task concerning this paper is an important baseline to try before moving to more complex methods is not novel. In any empirical field, more precisely in standard deep learning fields, practitioners are encouraged and often by default start with the worst baseline which is random. While I agree that the authors through this paper demonstrate the viability of random baseline in the domain of code generation, that doesn't negate the concern of the overarching novelty.
> >
> > Re: Implicit Knowledge Consensus - The authors state "This is also one reason we think Random works: if the pool already aggregates expert knowledge, we may not need an extra selection strategy on top of it." - This is extremely vague. While intuitive interpretation is helpful, in this case, there is no evidence (statistical or qualitative) to suggest that the interpretation is valid and hence the main claim of the paper becomes more abstract than the scientific requirements for a novel publication.
> >
> > Given these outstanding concerns, I remain at my original score.

---

> > > ### Author Response · Authors · 2026-04-05
> > >
> > > Thanks for helpful comments! We would like to summarize your questions and provide our responses below:
> > >
> > > 1. **Q1. how the contribution should be presented beyond Random**: Our claim is more specific: we study *within-problem solution selection* in a multi-solution code corpus. This setting also appears in RLVR pipelines, where reward-approved solutions can accumulate for one problem and later be reused in mid-training or SFT. Prior data-selection work mainly studies example- or domain-level selection, while our paper studies selection among many correct solutions to the same problem. The setting is large:
> > >
> > >    | Statistic | Value |
> > >    |-----------|------:|
> > >    | Raw solutions | 503,268 |
> > >    | Solutions after deduplication/cap | 154,714 |
> > >    | Mean solutions per problem, raw | 191 |
> > >    | Mean solutions per problem, after dedup/cap | 59 |
> > >
> > >    We will revise the title, abstract, and introduction accordingly. Specifically, our claim is that sophisticated proxy objectives do not yield stable gains in this setting. That is the claim we will adopt in revision.
> > >
> > > 2. **Q2. how the viability evidence contributes to the overall contribution**: We agree that viability alone would be incomplete. The contribution is the empirical characterization: complex selectors do not show a robust advantage, and their behavior varies with difficulty and preprocessing. So the paper should not rest on "Random works," but on when simple selection is enough and when complex methods may still help.
> > >
> > >    | Difficulty | Random Test Pass@1 | K-Means Test Pass@1 | Kernel Herding Test Pass@1 |
> > >    |------------|-------------------:|--------------------:|---------------------------:|
> > >    | Easy | **22.71** | 15.21 | 20.62 |
> > >    | Medium | 2.77 | 2.67 | **2.87** |
> > >    | Hard | 0.89 | 1.39 | **1.77** |
> > >
> > >    These results indicate a more specific pattern than "Random always wins." Cleaned, representative data already captures much of the benefit, while complex selectors show clearer gains mainly on the hardest tail. The preprocessing study points the same way: Random rises from **5.57** without deduplication to **6.32** with LSH deduplication plus cap, while K-Means rises from **4.47** to **4.87** and Kernel Herding from **5.44** to **6.23**. We will revise the paper so this empirical pattern, rather than a generic Random-baseline claim, is the center of the contribution.
> > >
> > > 3. **Q3. how "implicit knowledge consensus" should be presented more clearly**: We appreciate this suggestion. In revision, we will revise the title, abstract, and discussion so that the empirical claim remains central and the interpretation is stated more precisely. Our analyses already provide relevant signals: lower Pass@1 concentrates on harder and more diverse problems across methods, which is consistent with the possibility that diversity-seeking selection is partly entangled with difficulty in our setting.
> > >
> > >    | Difficulty | n | Levenshtein | Jaccard | Syntax | Pass@1 (%) |
> > >    |------------|--:|------------:|--------:|-------:|-------:|
> > >    | Easy | 48 | 88.3 | 0.42 | 0.69 | 23.1 |
> > >    | Medium | 101 | 179.9 | 0.48 | 0.75 | 2.8 |
> > >    | Hard | 79 | 229.0 | 0.51 | 0.78 | 0.9 |
> > >
> > >    These analyses guide this revision. The added case analysis agrees: lower-performing concordant cases are harder and more diverse than concordant successes, with average difficulty **2.37** versus **1.39** and average syntax diversity **0.76** versus **0.69**. We also report two direct statistical signals in the text: syntax diversity has Pearson correlation **-0.471** with Pass@1, with **p < 0.001**; and in the five-seed study Random reaches mean Test Pass@10 **18.07**, above **17.54** for Kernel Herding, **17.37** for Facility Location, and **17.19** for K-Means. We will use these analyses to sharpen the discussion and separate empirical observation from interpretation more clearly.
> > >
> > > We appreciate the reviewer's push to make the claim more precise. We think this is why the research is useful for practitioners working with RLVR-style or other verifiable corpora. Such pipelines with reward functions can easily produce more accepted traces than can be retained for later reuse in mid-training or SFT because storage, token budget, and training time remain limited after verification. Selection is therefore a practical necessity once accepted pools become large. Our contribution is not only to suggest Random as the default low-cost baseline and to clarify under what evidence more complex selection may become appropriate; it also includes broader empirical analyses of difficulty stratification, preprocessing effects, and diversity that indicate what kinds of analyses make that transition reasonable. Taken together, these observations provide a more complete methodology for this problem, and this methodology is itself part of what we view as the paper's contribution. We will revise the paper so this conclusion is clearer. We respectfully ask the reviewer to reconsider the score in light of this empirical value.

---

### Decision · Program_Chairs · 2026-04-30

**Decision:**

Accept (regular)

**Comment:**

This paper studies the problem of how to curate the training data when doing SFT for code generation. In particular, it looks at the problem of selecting one of the correct code from a large number of possible code solutions that are consistent with a given specification. Using extensive evaluation over four language models and six complex selection strategies, the paper shows that a simpler random selection technique is competitive and in some cases even better than more complex selection strategies. There were quite a few concerns that came up during the rebuttal process including questions around novelty of the findings, claims around applicability of findings, limited experimentation around Python-only scope and generalization to other benchmarks, effect of deduplication and correlation of difficulty with selection strategies, and more concrete learnings for the community. The rebuttal response with additional experiments and discussions helped with some of the concerns, while some other concerns still remained. I believe this paper still makes a good contribution to the community to show a simple random selection strategy works competitively compared to more complex strategies, and provides valuable guidance especially after including the new experiments in the response and additional discussions. It would be great to add them in the final version of the paper.